# Clinically Interpretable Rule–Guided Preference Optimization in Vision–Language Models for Radiology Report Generation

## Abstract

In modern healthcare, radiology plays a pivotal role in diagnosing and managing diseases. However, the complexity of medical image data combined with the variability of natural language generation often leads to inconsistencies, hallucinations, and a lack of clinical grounding, especially in automatically generated radiology reports. To address these challenges, we introduce a clinically interpretable rule-guided extension of direct preference optimization, tailored for radiology report generation. A typical radiology report comprises of findings and impression, findings capture the complex visual information from the medical image, for example a chest X-ray, and the impression is the implied conclusion. Our framework leverages on this phenomenon to design clinical rules from existing findings and impressions, that connect the finding and impression as a horn rule. The rules act as an additional, interpretable supervision signal, guiding the preference optimization of Vision–Language Models (VLM) toward outputs that are not only fluent but also clinically faithful. Unlike conventional preference optimization, which relies solely on lexical preferences, our approach enforces alignment with clinically meaningful predicates such as the presence, absence, or severity of key findings. A central feature of this framework is its ability to inject clinical rule guidance during optimization, ensuring that generated reports remain both linguistically coherent and clinically accurate. By integrating a neural verifier trained to evaluate rule satisfaction, our method provides an explicit mechanism for grounding preferences in interpretable clinical semantics via the clinical rules. Experimental results on benchmark datasets like MIMIC–CXR-JPG and IU–Xray, demonstrate that our approach substantially improves factual accuracy, and overall report quality compared to zero-shot and standard DPO baselines. We record a performance boost of $10\%$ and $9\%$ across lexical and semantic metrics. These results highlight the promise of clinically interpretable preference optimization as a pathway toward trustworthy radiology report generation in medical AI.

## 1 Introduction

Medical AI research has emerged as a promising direction in recent times Tăuțan et al. (2021); Xia et al. (2024); Tu et al. (2024); Hu et al. (2024). Recent progress in Vision-Language Models (VLMs) have pushed the boundaries on how machines align information across modalities, showing substantial performance in tasks involving image-text pairs Liu et al. (2023). Like Large Language Models (LLMs), VLMs are trained on massive image-text pairs, which imparts generalization capabilities. Following this Medical VLMs (Med-VLMs) are being developed that demonstrate striking performance in various use-cases from Medical Visual Question Answering to Radiology Report Generation Zhang et al. (2023); Wu et al. (2023); Moor et al. (2023); Sellergren et al. (2025); Wang et al. (2022b). However, these models suffer from inadequate factual grounding and misalignment issues Zhou et al. (2024); Sun et al. (2024). As a result, Med-VLMs often hallucinate, which appear to be coherent, but are unable to capture the corresponding information in medical image. To address this issue, several attempts have been made to utilise preference optimization for better image-text alignment and factual grounding in Med-VLMs Hein et al. (2024); Banerjee et al. (2024); Zhu et al. (2025). However, these methods heavily rely on safely curated preferred data,

without taking into account the clinical relevance. As a result, the preferred and disprefered pairs become easily distinguishable. This makes the model distribution skewed to learn only the lexical outline disregarding clinical semantics which is essential in a high stake medical AI system that should be able to generate fluent text which are grounded in clinical semantics. This skewed distribution leads to mis-diagnosis, missing pathology and hallucinated wrong measurements completely misaligned with the input X-ray, rendering the methods untrustworthy.

Towards this we observe that we can represent a particular medical image-text pair, which in our case is an chest X-ray image and its corresponding report, as an implication rule. We first extract entities and the corresponding modifiers from the findings and impressions of a report. A predicate rule is created based on the extracted modifier-entity pairs. Following, this we train a neural verifier, that checks whether a rule and the corresponding report are aligned or not. The preference dataset is then curated by performing multiple sampling of the VLM and ranking the outputs using the verifier and choosing the two topmost ranked outputs as preferred and dispreferred pairs. Doing this embeds the clinical relevance in the preference dataset curation. Our curation method captures the underlying clinical semantics, which when adhered to reduces the skewed behavior. This makes the finetuned models trustworthy for inclusion in clinical workflows. Our preference data curation technique along with the designed symbolic rules fills the gap of explicit clinical knowledge integration. With the aforementioned resources in our hand we design an optimization framework that utilizes the symbolic clinical rules as interpretable supervision, along with the preference data. We put forth a joint optimization method that fine-tunes VLMs to generate coherent and clinically guided outputs.

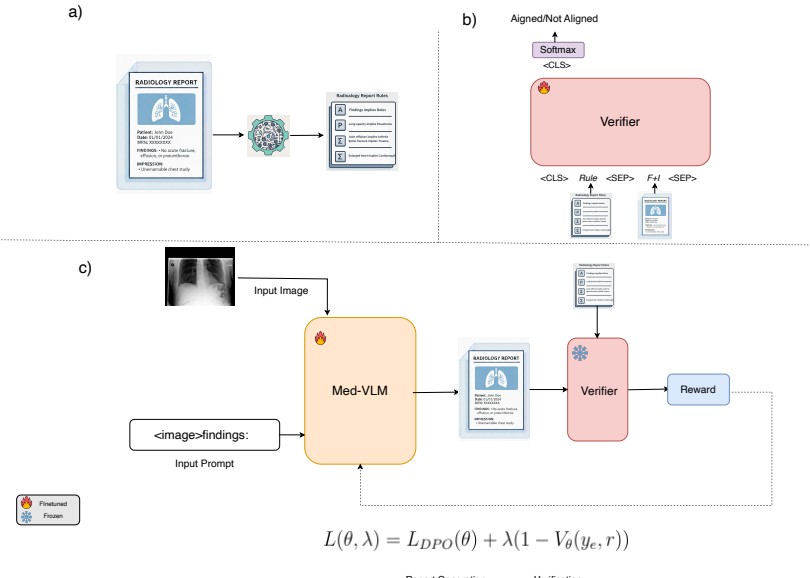

Figure 1: (a) For each report we extract the horn rules. (b) A Roberta based Neural Verifier is trained to verify a radiology report and a horn-rule, that outputs aligned/misaligned. (c) At each step the joint optimization process inputs a chest x-ray and a prompt to the VLM that produces the report which is verified and the verifier gives a score between 0-1. Finally, the proposed loss function optimizes the VLM to generate fluent and clinically grounded radiology reports.

Our contributions are:

1. A Clinically interpretable rule guided Radiology Report Generation Framework: Our method surpasses the existing frameworks by $10\%$ and $9\%$ across lexical and semantic metrics.

2. A neural rule verifier to assess the logical consistency between radiology reports and predicate rules. Beyond serving as a reward model, the verifier provides interpretable rule-based

assessment. The inclusion of this module in the generation pipeline gives 10% and 9% across lexical and semantic metrics.

3. A novel Rule-guided Direct Preference Optimization (DPO) framework, where the rule verifier score is used as a reward signal to fine-tune a multimodal VLM for report generation. This encourages outputs that are fluent as well as clinically grounded. Our approach outperforms the current frameworks by 10% and 9% across lexical and semantic metrics.

## 2 BACKGROUND

### 2.1 RADIOLOGY REPORT GENERATION VIA MEDICAL VISION-LANGUAGE MODELS

Medical vision-language models (VLMs) leverage transformer-based architectures and multimodal pretraining on medical images and reports for tasks such as radiology report generation and visual question answering (Wang et al., 2022b; Sellergren et al., 2025; Moor et al., 2023). Radiology report generation commonly uses encoder-decoder architectures (Vinyals et al., 2014; Xu et al., 2015; Pan et al., 2020), with improvements including image-text matching, hierarchical LSTMs, and memory modules for long-text decoding (Wang et al., 2021; Chen et al., 2020; Wang et al., 2022a). To mitigate data bias, some approaches integrate external knowledge via knowledge graphs, using graph neural networks, dynamic updates, or knowledge distillation (Li et al., 2019; 2023; Huang et al., 2023; Liu et al., 2021; Zhang et al., 2020; Kale et al., 2023).

### 2.2 CONSTRAINED PREFERENCE OPTIMIZATION

Constrained Direct Preference Optimization (Constrained DPO) Liu et al. (2024); Asadi et al. (2025); Yin et al. (2025),refines standard DPO methods for language model alignment by introducing explicit constraints on probability mass movement between preferred and dispreferred responses.

These constraints control the displacement between reference and learned policies, mitigating a known issue in vanilla DPO where both preferred and dispreferred responses can decrease in likelihood, potentially decreasing model helpfulness or safety. With monotonic constraint functions, C2-DPO optimizes the DPO objective and ensures robust, meaningful model behavior in safety-critical scenarios. Empirical studies show Constrained DPO provides consistent improvements in preference alignment and final model quality, outperforming vanilla DPO and other baselines on benchmark datasets.

Previous methods have primarily focused on demonstrating how carefully curated preference datasets can guide models toward human-aligned and trustworthy outputs. However, none of these approaches explicitly incorporate symbolic rules into the process which has been the center piece of our contribution.

## 3 METHODOLOGY

Our proposed method, as shown in Fig. 1, draws inspiration from the trustworthy alignment line of research Dai et al. (2024); Liu et al. (2024); Asadi et al. (2025). All of the previous methods explore how meticulously curated preference datasets can be useful for steering a model to human aligned and trustworthy outputs. However, none of the previous methods try to incorporate explicit symbolic rule for constraining the preference optimization. In a clinical setting reliance on explicit clinical rules gains more weightage over only human curated preference datasets. Recent VLM based works in medical NLP, particularly in radiology report generation, concentrate on maximizing fluency, coherency and capturing the reporting style Hein et al. (2024); Banerjee et al. (2024); Zhu et al. (2025). This frontier of research has led to generation of fluent and clinically styled text. However, in a sensitive domain like medical NLP, fluency alone do not suffice to accept and trust generated outputs. A degree of factual grounding in symbolic clinical knowledge is required, which in turn raises acceptability and increases trust. Nonetheless, all previous medical VLMs missout on this perspective. We aim to address this crucial aspect via safety-guided finetuning of medical VLMs with symbolic clinical rules. Therefore, our finetuning method enable the VLMs to generated fluent as well as clinically grounded radiology reports.

## 3.1 DATA PREPROCESSING

A radiology report consists of findings and impressions. An example of findings and impressions is given below.

---

**FINDINGS**: The cardiac silhouette is enlarged, consistent with cardiomegaly. Lung volumes are stable and remain low. No evidence of pneumothorax is seen. Minimal blunting of the right costophrenic angle is noted. No focal infiltrates are identified.
**IMPRESSIONS**: cardiomegaly with mild right pleural effusion.

---

We observe that each free-text radiology reports can be reduced to a rule structure, such as $findings \implies impression$. Findings, in this underlying structure comprises the exact clinical knowledge from the image, and impression contains the conclusion. This overall structure represents the manner in which a report is written in a real-world setting.

---

cardiomegaly and low lung volume and no pneumothorax and minimal right costphrenic blunting and no focal infiltrates

**implies**

cardiomegaly and mild right pleural effusion

---

Therefore, we include a pre-processing stage to reduce a free-text radiology into a natural language clinical rule. We connect each finding and impression via an implication, in the form of a predicate rule. Before training, we extract natural language predicate rules from each of the data samples. Following this stage, each dataset instance has a X-ray image, report and the corresponding natural language predicate rule connecting the findings and impression via an implication. The preprocessing pipeline consists of the following steps:

### 3.1.1 ENTITY EXTRACTION

Each report is parsed to identify radiological entities.

---

**Entities from Findings**: cardiomegaly, lung volume, pneumothorax, right costphrenic, infiltrates

**Entities from Impressions**: cardiomegaly, pleural effusion

---

To obtain structured clinical rules from free-text radiology reports, we employ the Stanza natural language processing toolkit Qi et al. (2020). Specifically, we use the MIMIC-trained English pipeline with the i2b2 Named Entity Recognition (NER) processor.

These extracted entities serve as the building block for the natural language predicate rule. Following, this we pair this with modifiers that signal the intensity of the disease in the provided chest x-ray. The extracted entities, clubbed with the modifiers, build the final predicate rule. The reduction of the unstructured radiology reports into structured rules, builds ground for grouding radiology report generation with domain-specific constraints.

### 3.1.2 MODIFIER DETECTION

Once entities are extracted, we perform contextual analysis to capture descriptive information that inform us about the instensity of the disease. These contextual cues are referred to as **modifiers**.

---

**Modifiers from Findings**: low, no, minimal

**Modifiers from Impressions**: mild

---

Entities along with modifiers form entity-modifier pairs that are used as individual literals in the final rule.

> **Entity modifier pair from Findings**:cardiomegaly, low lung volume, no pneumothorax, minimal right costrophrenic infiltrates
>
> **Entity modifier pair from Impressions**: cardiomegaly and mild pleural effusion

To identify these modifiers we first do a contextual analysis of the dataset and create a set of modifiers. With each radiology report we first extract the entity and search within a window of ten forward and backward tokens to find modifiers that map to our set of modifiers. By combining these approaches, the algorithm 2 produces structured entity–modifier pairs, which are later concatenated to form a predicate rule in natural language.

### 3.1.3   HORN RULES FORMATION

Following entity extraction and modifier detection, each entity–modifier pair forms the individual literal of the final rule. Therefore, the final horn rule captures the underlying nuanced clinical reasoning required for generating a fluent and clinically grounded radiology report. Throughout this process we make sure our method adheres to natural language rules.

Finally, a natural language Horn rule is formed that takes the following structure:

$$p_1 \wedge p_2 \wedge \cdots \wedge p_k \ \rightarrow \ q,$$

> cardiomegaly $\wedge$ low lung volume $\wedge$ no pneumothorax $\wedge$ minimal right costrophrenic infiltrates $\rightarrow$ cardiomegaly $\wedge$ mild pleural effusion

where the conjunction of predicates $(p_1, p_2, \ldots, p_k)$ represents evidence extracted from the findings of the radiology report, and $q$ denotes a entity-modifier pair from the impression. The entity-modifier pairs capture the disease and the intensity and represent it in the conjunction form, thus we chose the horn rule representation.

By systematically composing entity-level predicates into natural language Horn rule, we create an interpretable connection between raw textual descriptions and clinically relevant rules. This logical representation facilitates downstream reasoning, supports clinically grounded guidance of VLMs, which we further incorporate in our downstream task of radiology report generation.

### 3.2   CLINICAL RULE GUIDED VERIFIER

Our methodology for developing a verifier model is primarily inspired by the work of Clark et al. (2021), which established that transformer models can be trained to effectively reason over rules and paragraphs expressed in natural language.

### 3.2.1   NEURAL VERIFIER

We adopt this method and employ a RoBERTa-large architecture, framing the complex verification task as a straightforward binary classification problem. Given a radiology report $R$ (Findings: $F$, Impression: $I$) and a Horn rule $h$, the model predicts whether the report is *Aligned* or *Not Aligned* with the rule. To achieve this, we first construct a large-scale dataset as described earlier. Next, we label each aligned sample as 1 and misaligned as 0.

The report and rule are concatenated into a structured input sequence of the form:

$$X = \texttt{<CLS>}\ h\ \texttt{[SEP]}\ R\ \texttt{[SEP]},$$

This sequence is tokenized and passed through RoBERTa:

$$z = \text{RoBERTa}_\theta(X) \in \mathbb{R}^d,$$

where $z$ is the pooled embedding corresponding to the `<CLS>` token. A linear classification head projects this embedding into a scalar logit:

$$\ell = Wz + b, \quad W \in \mathbb{R}^{1 \times d}, \; b \in \mathbb{R}.$$

Finally, a sigmoid function maps the logit into an alignment probability:

$$\hat{y} = \sigma(\ell) = P_\theta(y = 1 \mid R, h), \quad \hat{y} \in [0, 1].$$

We keep all symbolic representations in fluent natural language. This makes the rules readily human understandable and enables us to move from a symbolic verifier to a neural verifier. RoBERTa model is fine-tuned on this dataset using a BCE loss function, with accuracy serving as the primary evaluation metric, justified by the balanced class distribution in our data.

### 3.2.2 NEURAL VERIFIER TRAINING

We prepare a dataset with aligned and misaligned samples based on the aforementioned rule extraction strategy. The details of aligned/misaligned data preparation is mentioned in Appendix A.1.1. Following, this the neural verifier is trained to classify whether a given report-rule pair is aligned/misaligned. We use the following loss function to train the neural verifier: **Neural Verifier Loss** ($L_V$): The neural verifier is trained to predict whether a radiology report $r$ is logically consistent with a set of rules $R$. We use a binary cross-entropy (BCE) loss, which penalizes the verifier when its predicted probability diverges from the ground-truth label $y \in \{0, 1\}$:

---

**Algorithm 1:** DPO with Neural Verifier for Clinically Interpretable Rule-guided Report Generation

**Input:** $\mathcal{D} = \{x_v^{(i)}, x_t^{(i)}, r^{(i)}\}_{i=1}^N$: Dataset; $x_p$: Prompt; $\pi_\theta(\cdot, \cdot)$: Med-VLM; $\mathcal{V}(\cdot, \cdot)$: Neural Verifier; $\epsilon$ : verifier threshold.
**Output:** Optimized model parameters $\theta^\star$ for Med-VLM.
Initialize empty preference dataset $\mathcal{D}_P$
**foreach** $(x_v, x_t) \in \mathcal{D}$ **do**
  ▷ *Generate candidate reports*
  Generate multiple responses: $r_m \leftarrow \pi_\theta(x_v, x_p)$
  **foreach** $r_i \in r_m$ **do**
    ▷ *Compute verifier score*
    Compute logical consistency: $v_i \leftarrow \mathcal{V}(r_i, x_t)$
  ▷ *Choose top two ranked responses*
  Preferred Response:
  $y_w \leftarrow$ Highest Verifier Score
  $y_l \leftarrow$ Second highest Verifier Score
  ▷ *Collect preference and verifier info*
  Add $(x_v, x_t, y_w, y_l, r)$ to $\mathcal{D}_P$
▷ *DPO + Verifier Constrained Optimization*
**foreach** $(x_v, x_t, y_w, y_l, r) \in \mathcal{D}_P$ **do**
  Compute DPO loss:
    $L_{\text{DPO}} \leftarrow -\log \sigma\big(\pi_\theta(x, y_w) - \pi_\theta(x, y_l)\big)$
  Compute Verifier loss: $L_V \leftarrow |1 - v|$
  Compute Lagrangian multiplier: $\lambda \leftarrow \text{softplus}(\lambda_{\text{param}})$
  Compute joint loss: $L_{\text{joint}} \leftarrow L_{\text{DPO}} + \lambda L_V$
  Backpropagate $L_{\text{joint}}$ and update $\theta$
  Backpropagate $-\lambda L_V$ and update $\lambda$
**return** $\theta^\star \leftarrow$ *optimized Med-VLM parameters*

---

$$L_V = \mathbb{E}_{(r,R,y)\sim D_{rule}} \left[ y \cdot \log p(y = 1 \mid r, R) + (1 - y) \cdot \log p(y = 0 \mid r, R) \right] \tag{1}$$

This objective ensures that the verifier learns to assign high confidence to logically consistent report–rule pairs while suppressing inconsistent ones. Here, $D_{rule}$ is the verification dataset, and $y$ is the ground-truth label for a given report $r$ and rule $R$. The verifier is first trained via this loss mentioned in 1.

### 3.3 INTERPRETABLE SUPERVISION VIA RULE-GUIDED JOINT OPTIMIZATION

While the verifier model ensures logical consistency, the quality of a generated report also depends on stylistic factors such as clarity, tone, and conciseness. To align the model's outputs with these nuanced clinically relevant linguistic preferences, we employ Direct Preference Optimization (DPO). This technique fine-tunes the generative model directly on a dataset of preferences, bypassing the need for an explicit reward model. However, in a clinical setting along with the stylistic factors, factuality is of paramount importance towards clinically grounded outputs. In this optimization setting the clinical guidance is ensured from the external interpretable supervision provided by the clinical

rule verifier The training process relies on a preference dataset, $D_P$, which contains triplets of the form $(R, y_w, y_l)$. For a given rule $R, y_w$, is the "winning" report (preferred) and $y_l$ is the "losing" report (dispreferred). The DPO loss function is formulated to increase the likelihood of the model generating the winning report over the losing one.

To ensure that the report generation model is both linguistically fluent and clinically grounded, we formulate a joint optimization problem. Specifically, the objective is to minimize the DPO loss, which promotes stylistic fluency and clinical reporting style, while simultaneously encouraging the verifier output to be as close to 1, indicating strong logical consistency with the provided clinical rules. We cast this as a constrained optimization problem and solve it via a Lagrangian formulation.

$$\theta^\star = \underset{\theta}{\arg\min}\, \mathbb{E}_{(x,y_w,y_l)\sim\mathcal{D}_\mathcal{P}}\Big[ -\log\sigma\big(\pi_\theta(x,y_w) - \pi_\theta(x,y_l)\big)\Big] \quad \triangleleft \textbf{DPO Loss}$$

$$\text{s.t. } \mathbb{E}_{(y_e,r)\sim\mathcal{D}_\mathcal{P}}\big[1 - v_\theta(y_e, r)\big] \leq \varepsilon \qquad\qquad \triangleleft \textbf{Neural Verifier Constraint}$$

(2)

The primary components are:

**1. The DPO Loss ($L_{DPO}$):** This loss aligns the model with fluency preferences by maximizing the likelihood of generating a "winning" report $y_w$ over a "losing" one $y_l$.

$$L_{DPO} = \mathbb{E}_{(x,y_w,y_l)\sim D_P}\left[ -\log\sigma\left(\beta\log\frac{\pi_\theta(y_w|x)}{\pi_{ref}(y_w|x)} - \beta\log\frac{\pi_\theta(y_l|x)}{\pi_{ref}(y_l|x)}\right)\right] \qquad (3)$$

**2. The Rule Loss ($V(x, r)$):** This loss acts as the interpretable supervision which constraints the primary VLM to generate fluent and clinically consistent outputs, by pushing the med-VLM to generate reports that align properly to its corresponding rule.

$$V_\theta(y_e, r) = (\mathbb{E}_{(x,y_e)\sim D_P}\left[(1 - Verifier(\pi_\theta(y_e|x), r))\right] \qquad (4)$$

**The Joint Loss Function:** We combine these objectives into a single Lagrangian, where we minimize the DPO loss subject to the constraint $1 - V_\theta(y_e, r)$ very close to $\varepsilon$, $\varepsilon$ is a very small number near to 0, $\varepsilon \approx 0^+$.

$$L(\theta, \lambda) = L_{DPO}(\theta) + \lambda(1 - V_\theta(y_e, r)) \qquad (5)$$

In this formulation:

- $\theta$ represents the parameters of $\pi_\theta$.
- $\lambda \geq 0$ is the Lagrange multiplier, which acts as a dynamic penalty weight.
- $\varepsilon$ is a hyperparameter that defines the acceptable upper bound for the verifier loss.
- $y_e$ represent the report generated by $\pi_\theta$ at each epoch.

During training, $\lambda$ is updated dynamically. If the verifier loss exceeds the threshold ($L_V > \epsilon$), $\lambda$ increases, forcing the model to prioritize logical consistency. If the constraint is satisfied ($L_V \leq \epsilon$), $\lambda$ decreases, allowing the optimization to focus more on improving the report quality via the DPO objective. This approach provides an adaptive and principled way to balance the trade-off between generating high-quality reports and ensuring they remain factually correct. Algorithm 1 demonstrates the our workflow, we also provide the gradient update expressions in Appendix 3.

## 4 EXPERIMENTAL SETUP

### 4.1 DATASET

We evaluate our approach on two widely used public radiology benchmarks: MIMIC-CXR-JPG Johnson et al. (2019) and IU-Xray Demner-Fushman et al. (2015). MIMIC-CXR-JPG consists of approximately 377,000 frontal and lateral chest X–ray images paired with free-text radiology reports, providing a large-scale setting for multimodal learning. IU-Xray contains around 7,000 studies, each with corresponding chest X–ray images and reports, and is commonly used to evaluate generalization and cross-dataset performance.

For both datasets, we follow a consistent preprocessing pipeline as aforementioned. The datasets are split into training, validation, and test sets following the recommended splits, ensuring no patient

| Models | IUX | | | | | MIMIC-CXR-JPG | | | | |
|---|---|---|---|---|---|---|---|---|---|---|
| | BL-1 | BL-2 | BL-3 | BL-4 | RG-L | BL-1 | BL-2 | BL-3 | BL-4 | RG-L |
| Med-Gemma (ZS) | 27.39 | 14.72 | 07.19 | 02.33 | 24.17 | 15.17 | 7.66 | 3.64 | 1.50 | 18.53 |
| + DPO | 25.17 | 14.27 | 07.48 | 02.73 | 24.09 | 16.15 | 08.08 | 03.97 | 01.84 | 17.70 |
| +DPO+Ver (Ours) | 29.36 | 17.48 | 09.97 | 05.39 | 27.71 | 22.90 | 11.38 | 05.66 | 02.64 | 18.89 |
| Med-Flamingo (ZS) | 17.55 | 08.44 | 01.27 | 0.11 | 15.36 | 12.49 | 05.75 | 01.93 | 00.33 | 15.64 |
| +DPO | 17.83 | 08.39 | 02.53 | 00.69 | 18.77 | 20.04 | 07.77 | 03.31 | 00.81 | 16.77 |
| +DPO+Ver (Ours) | 18.41 | 09.35 | 03.06 | 00.55 | 19.22 | 22.86 | 09.91 | 01.87 | 00.45 | 18.45 |
| Gemma3n (ZS) | 11.16 | 05.80 | 01.68 | 00.33 | 13.80 | 10.35 | 04.76 | 01.45 | 00.23 | 11.55 |
| +DPO | 12.81 | 05.53 | 02.31 | 00.61 | 15.45 | 11.85 | 05.12 | 02.45 | 00.33 | 11.96 |
| +DPO+Ver (Ours) | 13.53 | 06.19 | 01.74 | 00.41 | 16.48 | 12.25 | 05.65 | 01.88 | 00.45 | 11.45 |

Table 1: Comparison of VLMs for radiology report generation across multiple evaluation metrics, including BLEU, and ROUGE score for assessing the lexical accuracy. The table highlights the improvements achieved by our DPO+Verifier framework over zero-shot and standard DPO baselines.

overlap across sets. This standardized preprocessing ensures that both textual and visual modalities are compatible with our multimodal model, allowing for robust evaluation of linguistic quality and clinical consistency in generated reports. The dataset preprocessing details along with splits are provided in Appendix A.1. We also provide the experimental setup in Appendix A.5.

## 5 RESULTS

We perform evaluations for three types of models. First, model pretrained on image-report pairs, MedVQA data and general pretrained.

### 5.1 GLOBAL ASSESSMENT

In this section we present the comparison of our proposed clinically interpretable rule-guided preference optimization against DPO and zero shot baselines.

#### 5.1.1 LEXICAL ANALYSIS

In this section we evaluate our proposed method on traditional lexical metrics like BLEU Papineni et al. (2002), Rouge-Score Lin (2004) scores. In table 5.1.1, the metrics provide a convenient means of measuring word overlap and syntactic similarity between generated and reference texts. Our method, DPO+Ver demonstrates a considerable increase in lexical scores. On comparing, DPO with Zero-Shot (ZS) baselines, as shown in Table 5.1.1, training with a DPO objective pushes the VLM towards memorizing the lexical structure. In some cases over memorizing as shown in 5.1.1. This leads to outputs which are lexically similar but misses out on key medical termiologies. In the DPO+Ver setting we see the model is better at generating outputs which are linguistically coherent and clinically grounded. The improved lexical metric score reinforces the output utility of the constrained optimization setup, where the neural rule verifier always prevents the model distribution to get skewed towards linguistic coherence alone. The highlighted scores show the efficacy of our method. However, lexical metrics are limited in the medical domain because they fail to capture semantic and contextual accuracy. For instance, *"There is focal consolidation"* and *"There is no focal consolidation"* are lexically similar but semantically opposite, highlighting the need for semantic-based evaluation in medical text generation.

#### 5.1.2 SEMANTIC ANALYSIS

We further evaluate our models using clinical semantic metrics, like ClinicalBERTScore Shor et al. (2023), RadGraph-F1 Jain et al. (2021), GREEN Ostmeier et al. (2024). Further, we utilize LLM based evaluation using DeepSeek-R1-Distill-Qwen-14B DeepSeek-AI et al. (2025). These metrics

| Models | IUX | | | | MIMIC-CXR-JPG | | | |
|---|---|---|---|---|---|---|---|---|
| | CBS | RG-F1 | LLM | GREEN | CBS | RG-F1 | LLM | GREEN |
| Med-Gemma (ZS) | 87.69 | 26.02 | 03.68 | 00.40 | 87.12 | 16.17 | 03.09 | 00.20 |
| + DPO | 85.68 | 28.00 | 03.65 | 00.45 | 86.79 | 15.55 | 03.35 | 00.22 |
| +DPO+Ver (Ours) | 88.17 | 31.00 | 03.85 | 00.48 | 90.39 | 18.18 | 03.25 | 00.25 |
| Med-Flamingo (ZS) | 86.44 | 16.64 | 03.25 | 00.38 | 80.55 | 08.45 | 03.12 | 00.23 |
| +DPO | 88.35 | 19.22 | 03.55 | 00.42 | 86.61 | 10.47 | 03.23 | 00.27 |
| +DPO+Ver (Ours) | 89.55 | 20.26 | 03.88 | 00.51 | 88.45 | 13.72 | 03.45 | 00.31 |
| Gemma3n (ZS) | 80.55 | 07.33 | 02.57 | 00.28 | 77.75 | 08.55 | 2.85 | 00.18 |
| +DPO | 86.66 | 10.72 | 02.67 | 00.33 | 79.75 | 08.75 | 02.88 | 00.23 |
| +DPO+Ver (Ours) | 88.45 | 13.73 | 02.98 | 00.37 | 80.11 | 08.55 | 03.11 | 00.27 |

Table 2: Comparison of VLMs for radiology report generation across multiple evaluation metrics, including ClinicalBERTScore (CBS), GREEN, LLM-based, and RadGraph-F1 (RG-F1) for assessing the accuracy of clinically relevant relational structures. The table highlights the improvements achieved by our DPO+Verifier framework over zero-shot and standard DPO baselines.

are particularly important for radiology reporting, where subtle differences such as "no pleural effusion" versus "pleural effusion present" can fundamentally alter diagnostic meaning, maintaining similar lexical structure.

For the demonstrated VLMs, we see a general trend of increment in performance in semantic metrics too. Specifically, in metrics such as RG-F1 and GREEN, check whether the radiological entities are intact in the generated text similar to the ground truth. Our proposed method show a considerable increment in these scores across models and benchmarks. This certifies, our method is efficient at maintaining the clinical entities in generated text. This attribute is owed to the external interpretable supervision signal that comes from the verifier. Similarly, for CBS and LLM (Appendix A.2 contains the llm prompt) scores which compares the overall fluency of the language, increment in these also show the utility of the constrained objective. Appendix A.3 shows the failure cases as well.

Taken together, lexical and semantic results demonstrate that while large vision-language models can achieve strong terminology coverage, their relational reasoning remains limited in the absence of fine-tuning. Direct Preference Optimization helps address this gap but lacks clinical grounding and entity preservation when applied alone. The integration of a neural verifier into the optimization loop strikes a better balance, allowing the model to preserve linguistic coherence as well as keeping the generation clinically grounded, which maintains clinical trustworthiness.

## 5.2 NUANCED ASSESSMENT

Direct Preference Optimization with a neural verifier (DPO+Ver) enhances medical vision–language models (medVLMs) by aligning outputs with clinically meaningful criteria. In-task gains show improved behavior on the training distribution, while cross-task improvements indicate generalization to unseen data, critical for real-world deployment under distribution shifts. Unlike general-purpose VLMs, MedVLMs inherently generate at least one clinically valid output due to domain priors, making reinforcement-style optimization feasible. The verifier identifies correct candidates, and the optimization amplifies this behavior across outputs, ensuring generated reports remain both linguistically coherent and clinically accurate, effectively propagating safe and reliable clinical reasoning throughout the model's predictions.

**Rule guided preference optimization of in task pre-trained Med-VLMs**: The approach begins by fine-tuning Med-Gemma that already achieves strong baseline performance ensuring that reinforcement-style alignment is feasible. Rule-based preferences are then integrated, where the verifier assigns higher scores to outputs consistent with medical logic (e.g., preferring "no pneu-

mothorax" over a hallucinated condition), and DPO propagates these preferences. This mechanism improves in-task accuracy (Radiology Report Generation) by up to $10\%$.

**Rule guided preference optimization of cross task pre-trained Med-VLMs**: While DPO improves alignment within the training domain, it often struggles to transfer effectively across tasks Yan et al. (2025). By integrating a neural verifier grounded in domain-specific rules, our method (DPO+Ver) consistently reinforces clinically valid outputs while suppressing hallucinations, leading to improved performance beyond the training task (Med-Flamingo is trained on VQA but shows significant performance even with Radiology Report Generation). This demonstrates that preference optimization guided by medical rules not only strengthens trustworthiness in the source domain but also instills generalizable clinical reasoning, making cross-task pretrained Med-VLMs more robust and trustworthy for real-world medical applications.

**Rule guided preference optimization of general purpose pretrained Non Medical VLMs**: We extend rule-guided preference optimization to general-purpose pretrained vision–language models (VLMs) that lack domain-specific medical knowledge. Unlike Med-VLMs, these models often fail to generate even a single clinically valid output consistently, which limits the effectiveness of preference-based alignment. However, we do a supervised fine-tuning before applying our method. By integrating structured rules with a neural verifier, our approach encourages the model to prioritize outputs that satisfy desired criteria and reduces hallucinations. However, the performance boost is relatively modest compared to Med-VLMs, reflecting the challenge of applying rule-guided preference optimization when the base model does not possess strong domain priors. The numbers itself are moderate, therefore increment is also less impactful. This highlights that while the method is broadly applicable, its impact depends heavily on the underlying model's familiarity with the target domain.

## 6 CONCLUSION

In this work, we presented a clinically interpretable rule-guided extension of Direct Preference Optimization for radiology report generation. By leveraging the natural structure of radiology reports—findings and impressions—we designed Horn-rule–based constraints that connect visual evidence with clinical conclusions. This framework introduces interpretable supervision signals and integrates a neural verifier to enforce rule satisfaction, ensuring generated reports remain both linguistically coherent and clinically faithful. Our experiments on MIMIC–CXR-JPG and IU–Xray demonstrate substantial improvements in factual accuracy and overall report quality, with gains of $10\%$ and $9\%$ across lexical and semantic metrics. These results underscore the potential of clinically interpretable rule-guided preference optimization as a reliable pathway for developing trustworthy medical AI systems.

## ETHICS STATEMENT

Our proposed framework enables VLMs to generate outputs which are linguistically fluent and clinically grounded. All medical data is anonymized, and our data processing pipeline prevents any risk of patient identity leakage. The framework is designed to assist, not replace, clinicians by enhancing diagnostic accuracy while promoting interpretable and trustworthy outputs. Bias is mitigated through diverse training data covering various demographics and medical conditions, and the neural verifier ensures that generated reports adhere to clinical rules. Patient-facing explanations are clear, clinically accurate, and free from misleading information. Human oversight guarantees that outputs maintain clinical validity, ethical standards, and fairness in medical AI applications.

## STATEMENT OF LARGE LANGUAGE MODEL USAGE

Large language models (LLMs) were employed to assist in drafting text, generating prompts, and performing semantic evaluations. All outputs from the LLMs were carefully reviewed and curated by the authors to ensure accuracy, consistency, and adherence to clinical standards

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

# A APPENDIX

## A.1 DATA PREPARATION

---

**Algorithm 2:** Radiology Report → Horn Rule

---

**Input: Input:** $R$:Radiology report
**Output: Output:** Horn rule $h$
Extract entities $\mathcal{E}$ from $R$ using Stanza i2b2 NER
Initialize empty set of predicates $\mathcal{P}$
**foreach do**
  $\llcorner$ each entity $e \in \mathcal{E}$
Search within $\pm 10$ tokens around $e$ in $R$
Identify modifiers $m \in \{$Negation, Intensity, Location$\}$
Form entity–modifier pair $(e, m)$
Convert $(e, m)$ into predicate $p = \texttt{Predicate}(e, m)$
Add $p$ to $\mathcal{P}$
Initialize impression as clinical conclusion $q$ from predicates $\mathcal{P}$
Construct Horn rule: $h = p_1 \wedge p_2 \wedge \cdots \wedge p_k \;\; \rightarrow \;\; q$
**return** $h$

---

### A.1.1 REPORT-RULE DATASET

Following the rule extraction procedure described in Algorithm 2, each radiology report is paired with its corresponding clinical rule and labeled as aligned (label 1), indicating that the report satisfies the logical relationship encoded by the rule. To create negative examples, we generate dispreferred or altered versions of the same report that violate the rule, labeling these as misaligned (label 0). This process results in a dataset of aligned and misaligned report–rule pairs, providing explicit supervision for training the neural verifier to distinguish clinically consistent outputs from inconsistent ones, thereby grounding the model in interpretable clinical logic.

### A.1.2 PREFERENCE DATA CURATION

For preference optimization to be effective, the underlying pre-trained model must be capable of generating at least one output that can be considered a preferred sample. Building on this principle, we adopt a sampling-based strategy: for each model under consideration, we generate multiple candidate reports and then employ a pre-trained verifier to score and rank them. From this ranked list, we select the two highest-scoring candidates, designating the top-scoring report as the *preferred* sample and the second-highest as the *dispreferred* sample.

In terms of dataset preprocessing, we apply this procedure comprehensively across the entire IU-X dataset, extracting structured rules for all samples to support rule-guided supervision. For the MIMIC-CXR-JPG dataset, due to its large scale, we restrict this process to a randomly selected subset of 10,000 samples while maintaining the same rule-extraction pipeline. This design choice is justified by the fact that both Med-VLMs used in our experiments have been previously exposed to large-scale medical datasets, which facilitates generalization even when subsampling.

For general-purpose pre-trained vision–language models that have not been trained on medical data, we perform an additional supervised fine-tuning (SFT) step on paired image–text data before applying preference optimization. This ensures that the model develops a minimal level of domain alignment.

Apart from that, we use the official test split for both the datasets.

## A.2 LLM-Prompt

Your evaluation must adhere to the following sub-dimensions of similarity:

1. **Factual Equivalence:** Do both reports describe the same medical findings?
2. **Completeness:** Does the generated report capture all key information from the ground truth?
3. **Absence of Contradiction:** Does the generated report introduce information that conflicts with the ground truth?

**Reports:**

- Ground Truth Report: {ground_truth_text}
- Generated Report: {vlm_output_text}

**Likert Scale for Similarity:**

- 5: Strongly Agree
- 4: Agree
- 3: Neutral
- 2: Disagree
- 1: Strongly Disagree

**Task:** Provide a JSON object with the evaluation:

```
{
  "similarity_score": <1-5>,
  "rationale": "<brief justification>"
}
```

## A.3 Generated Outputs

**Zero-Shot**: findings:heart size, mediastinal and hilar contours are normal. lungs are clear. no pleural effusion or acute skeletal abnormality. impression: no radiographic evidence of pneumonia.
**DPO**: findings: heart size, mediastinal and hilar contours are normal. lungs and pleural surfaces are clear. mild s-shaped scoliosis is noted. impression: no radiographic evidence of pneumonia. impression: No acute findings.
**DPO+Ver**: findings: heart size, mediastinal and hilar contours are normal. lungs and pleural surfaces are clear. scoliosis is demonstrated. impression: no acute cardiopulmonary radiographic abnormality.This is a chest x-ray. The heart is normal size, the mediastinum is normal, and the lungs are clear.

The evaluation of the candidate report reveals that it includes a single false finding, specifically mild midthoracic dextroscoliosis, which was incorrectly reported. Apart from this, the report does not omit any findings present in the reference, misidentify anatomical locations, or misassess the severity of findings, indicating strong overall accuracy in capturing the reported observations. There are also no instances of mentioning comparisons that are absent in the reference, reflecting careful adherence to the source content. However, the report does omit a comparison detailing changes from a prior study, which could be clinically relevant for tracking patient progression. Overall, while the report demonstrates high fidelity to the reference, and shows the efficacy of our method to capture the linguistic reporting style along with maintaining clinical grounding, by keeping the entities intact.

**Generated**: findings: small right apical pneumothorax. multiple right-sided rib fractures are visualized. there is a small amount of pleural fluid tracking along the fissure. the lungs are clear. the cardiomediastinal silhouette is within normal limits. impression: small right apical pneumothorax, multiple right-sided rib fractures, and pleural fluid.
**Ground Truth**: findings:there are low lung volumes with associated bronchovascular crowding and basilar subsegmental atelectasis there is stable prominence of the right cardiac silhouette the mediastinum is unremarkable there is no pleural effusion, pneumothorax, or focal airspace disease there are stable chronic degenerative changes of the thoracic spine impression:no acute cardiopulmonary abnormality
**Generated**: findings: large right pneumothorax with basilar and apical components. there is no evidence of mediastinal shift. opacity in the right lung base is present, likely reflecting a combination of atelectasis and localized contusion. multiple right-sided rib fractures are demonstrated. small right pleural effusion is also demonstrated.
**Ground Truth**: findings:the heart is enlarged the mediastinal contours are stable large hiatal hernia is identified containing stomach and colon questionable small bowel loops are also visualized inthe large hernia there is adjacent compressive atelectasis the lungs are clear impression:no acute disease stable appearance of large hiatal hernia containing stomach and large bowel as well as possible small bowel loops.

## A.4 GRADIENT UPDATE

---

**Algorithm 3:** Training for DPO + Verifier Constraint

---

**Input:** Training dataset $\mathcal{D}$, learning rates $\eta_\theta, \eta_\lambda$
**Output:** Updated model parameters $\theta$ and multiplier parameter $\phi$
Initialize model parameters $\theta$ and multiplier parameter $\phi$
Compute initial multiplier: $\lambda = \text{softplus}(\phi)$
**foreach** *epoch* **do**
    **foreach** *minibatch* $(x_v, x_t, y_w, y_l, r) \in \mathcal{D}$ **do**
        Compute DPO loss:

$$L_{\text{DPO}} = -\log \sigma\big(\pi_\theta(x, y_w) - \pi_\theta(x, y_l)\big)$$

        Compute verifier loss:
$$L_V = |1 - v_\theta(x_v, r)|$$

        Compute current multiplier:
$$\lambda = \text{softplus}(\phi)$$

        Form primal loss:
$$L_{\text{primal}} = L_{\text{DPO}} + \lambda L_V$$

        Form dual loss:
$$L_{\text{dual}} = \lambda L_V$$

        **Primal update:**
$$\theta \leftarrow \theta - \eta_\theta\big(\nabla_\theta L_{\text{DPO}} + \lambda \nabla_\theta L_V\big)$$

        **Dual update:**
$$\phi \leftarrow \phi + \eta_\lambda\, L_V\, \sigma(\phi)$$

        Update multiplier:
$$\lambda = \text{softplus}(\phi)$$

**return** $\theta$, $\phi$

---

## A.5 EXPERIMENTAL SETUP

### A.5.1 NEURAL VERIFIER TRAINING

Table 3: Experimental setup for Neural Verifier Training.

| Component | Configuration |
|---|---|
| Base Model | RoBERTa-base ("Riiid/kda-roberta-base-race") |
| Classifier Head | 2-layer MLP + ReLU + Dropout, binary output |
| LoRA Config | $r = 8$, $\alpha = 16$, target modules = {query, key}, dropout=0.05 |
| Tokenizer | RobertaTokenizer, with padding |
| Dataset | RuleVerifierDataset (MIMIC, JSON with rule–report pairs) |
| Task | Binary classification (rule consistency) |
| Batch Size | 128 |
| Optimizer | AdamW, learning rate $1 \times 10^{-4}$, weight decay=0.01 |
| Scheduler | Cosine schedule, warmup ratio 5% |
| Loss Function | Cross-Entropy Loss |
| Epochs | 120 |
| Metrics | Precision, Recall, F1-score, Accuracy (micro-averaged) |
| Device | NVIDIA GPU (CUDA) |

### A.5.2 DPO TRAINING

Table 4: Experimental setup for DPO fine-tuning.

| Component | Configuration |
|---|---|
| Backbone Model | Med-Gemma (4B, multimodal, Image-Text-to-Text). |
| LoRA Configuration | Rank $r = 8$, $\alpha = 16$, dropout=0.1, applied to q_proj and v_proj. |
| Dataset | Indiana University Chest X-ray (paired with rule-annotated reports). JSON annotation with chosen vs. rejected samples. |
| Image Preprocessing | Resize 224×224, normalization mean=[0.5], std=[0.5], RGB conversion. |
| Tokenizer / Processor | HuggingFace AutoProcessor (Google Med-Gemma). |
| Loss Function | Direct Preference Optimization (DPO) loss with $\beta = 0.1$. |
| Optimizer | Adam, learning rate = $5 \times 10^{-4}$. |
| Scheduler | Linear scheduler with 5% warm-up steps. |
| Batching | Batch size = 2, gradient accumulation = 8 (effective batch = 16). |
| Precision | Mixed precision training with bfloat16 + gradient scaling. |
| Epochs | 20 epochs. Model checkpoint saved at epoch 20. |

### A.5.3 DPO+Verifier Joint Training

Table 5: Experimental setup for DPO + Verifier training.

| Component | Configuration |
|---|---|
| **Base Model** | Med-Gemma (conditional generation) with LoRA (rank = 8, $\alpha$ = 16, dropout = 0.1) applied to `q_proj` and `v_proj`. |
| **Verifier Model** | RoBERTa-base (Riiid/kda-roberta-base-race) with custom binary classifier head and LoRA (rank = 8, dropout = 0.05) on query/key projections. |
| **Dataset** | Indiana University Chest X-ray dataset; normalized PNGs with paired reports and predicate rules. |
| **Image Preprocessing** | Resize to 224×224, normalization to [-1, 1]. |
| **Optimization (Generator)** | Adam optimizer, learning rate = 5e-4, linear scheduler with 5% warm-up. |
| **Optimization (Verifier Constraint)** | Adam optimizer on Lagrange multiplier, learning rate = 4e-2. |
| **Batching** | Batch size = 2, gradient accumulation = 8 (effective batch = 8). |
| **Training Duration** | 15 epochs, checkpoint every 2 epochs. |
| **Precision** | Mixed precision with bfloat16 and gradient scaling. |

