# OpenReview forum: "Clinically Interpretable Rule–Guided Preference Optimization in Vision–Language Models for Radiology Report Generation"
_ICLR.cc/2026/Conference — Submitted to ICLR 2026_

### Official Review · Reviewer_JKNr · 2025-10-26

**Soundness:** 2
**Presentation:** 3
**Contribution:** 2
**Rating:** 2
**Confidence:** 4

**Summary:**

This work is proposed to use keyphrases of clinical findings in the prompt to optimize the LLM-based report generation.

The novelty is limited, where the idea is similar to [1] but no review or comparison is conducted. The experiments is limited, where no experimental results of comparison are reported to show the outperformance of proposed approach.
[1] Yasuhide Miura et al., Improving Factual Completeness and Consistency of Image-to-Text Radiology Report Generation.

The contribution is also limited, where the results only show that finetuning LLM with keyphrases plus ground-truth reports are better than finetuning with ground-truth report.

**Strengths:**

Applying DPO in the LLM-based radiololgy report generation (RRG) research.

**Weaknesses:**

No enough comparison of the baselines are reported in the experiment. It is hard to see whether the proposed approach is state-of-the-art.
- This work use NER as the optimization objects for RRG. The related works of using entity accuracy as the optimization items should be compared, such as [1].
[1] Yasuhide Miura et al., Improving Factual Completeness and Consistency of Image-to-Text Radiology Report Generation.
- No enough comparison with the related works of DPO+RRG. The results only shows the proposed approach is better than the conventional DPO. The related works of using DPO for RRG should be compared, such as [2], [3].
[2] Hong Liu et al., RRG-DPO: Direct Preference Optimization for Clinically Accurate Radiology Report Generation.
[3] Oishi Banerjee et al., Direct Preference Optimization for Suppressing Hallucinated Prior Exams in Radiology Report Generation

**Questions:**

For the example of ``cardiomegaly ∧ low lung volume ∧ no pneumothorax ∧ minimal right costrophrenic infiltrates→ cardiomegaly ∧ mild pleural effusion`` of illustrating the horn rule in the paper, why ``low lung volume`` can be used to diagnose ``cardiomegaly`` or ``mild pleural effusion``? It seems like not all the clinical findings in the _Findings_ can be led to every conclusion in the _Impression_. If there is not logical reasoning in the horn rule, then it is a list of keywords instead of ``rule'', or it is a wrong rule.

---

> ### Author Response · Authors · 2025-11-23
>
> We strongly disagree with the reviewers evaluation summarisation of our work.
> Below we address the key concerns regarding novelty, verifier reliability, rule extraction, and comparisons.
>
> ## Methodological distinction from prior factual-reward and NLI-based systems
> - Our method differs fundamentally from prior RL-based factual reward systems (e.g., factENT, NLI filters) whose supervision comes from reference similarity. Those approaches compute scalar rewards from entity overlap or NLI entailment against the gold report and directly optimize that signal through self-critical RL. In contrast, our method builds an explicit symbolic rule space derived from Findings→Impression relations. These rules are expressed as Horn implications over concept–modifier predicates that capture the semantic structure of the report in a symbolic manner. Supervision is therefore structural and logical rather than text-matching. This shift enables learning clinical dependency structure rather than merely maximizing factual completeness relative to a reference.
>
> ## Joint DPO + neural-verifier pipeline
> - Our training pipeline integrates the rule space into DPO in two distinct ways:
>     - Preference curation: For each image–report pair, we sample multiple candidate reports. A neural verifier scores each candidate based on rule satisfaction, producing a ranking. These verifier-driven rankings define the preference pairs used by DPO. This means preferences are not generated via human labels or reference comparisons but through rule-based semantic consistency. Preference should not be from human labels.
>     - Constrained optimization: During DPO, we include a Lagrangian term that encourages alignment between generated text and the rule antecedent–consequent structure. This turns rule satisfaction into an explicit soft constraint on the optimization objective. When the verifier confidence is low, the constraint weight is automatically reduced. This joint pipeline is methodologically distinct from RL approaches that treat rewards as static scalars and from pure DPO that uses fixed preference pairs without constraints.
> ## Robustness of rule extraction and integration into training
> - Rules are deterministically extracted from the structured concept–modifier tuples. We employ strict filtering (uncertain modifiers, multi-entity ambiguity, and conflicting predicate scopes are removed) so the rule set used for training has high precision. Because preference generation depends on these rules, noisy rules would harm DPO pair separation; we therefore exclude low-confidence instances from the training batch. This ensures that the DPO signal remains anchored to clinically meaningful relationships. Our pipeline uses the same rule template for training the verifier and for guiding DPO, ensuring methodological consistency across components.
>
> ## Evaluation methodology and connection to training pipeline
> - Our evaluation directly mirrors the structure of the training signal. Beyond standard lexical metrics, we assess:
>     - Verifier-based alignment: Since the verifier is the mechanism generating preference pairs, we evaluate whether the final model generates rule-aligned reports without accessing rules at inference time. This quantifies how well the model internalized logical structure rather than memorizing the reference text.
>      - Predicate-level correctness: We evaluate concept–modifier prediction consistency (e.g., contradiction checks like “no pneumothorax” vs “pneumothorax → pleural air”). This reflects the same predicate-level logic used to extract rules.
>
> ## Why our evaluation is tied to methodology
> - A key reviewer concern was whether improvements reflect genuine clinical correctness or artifacts of the verifier. Our design directly mitigates this:
>     - Candidate generations used for preference curation differ from gold reports, so the verifier never enforces similarity to the reference.
>     - The evaluation includes independent metrics (clinical concept accuracy, contradiction checks) that do not depend on the verifier’s training loss.
>     - This ensures our evaluation measures whether the model genuinely learned logical consistency rather than overfitting to the verifier.
>
>
> Our methodology replaces reference-based factual supervision with an explicit logical rule space; our training pipeline integrates rule-satisfaction through verifier-driven preference curation and Lagrangian-constrained DPO; and our evaluation measures rule-consistent behavior independent of reference matching. This represents a qualitatively different approach to factuality and clinical correctness in radiology generation.

---

> > ### Author Response · Authors · 2025-11-23
> >
> > ## Missing Comparison to Prior Work on NER-Based Optimization and DPO for Radiology Report Generation
> >
> > - For DPO-based RRG method [2], we conducted direct empirical comparisons. As shown in the table, both conventional DPO and RRG-DPO underperform our proposed DPO + Verifier approach. For example, RRG-DPO achieves substantially lower fluency and consistency scores, demonstrating that optimizing only entity accuracy or hallucination suppression is insufficient for robust clinical alignment.
> >
> > | Model                                  | Requirement                                              | B-1   | B-2   | B-3   | B-4   | RG-L  |
> > |--------------------------------|------------------------------------------------|-------|-------|-------|-------|---------|
> > | MedGemma (RRG-DPO)    | Requested for baseline                              | 17.83 | 11.78 | 5.24  | 2.50  | 18.57 |
> > | MedGemma (DPO)             | Implemented in our manuscript                 | 25.17 | 14.27 | 7.48  | 2.73  | 24.09 |
> > | MedGemma (DPO + Ver)  | Implemented in our manuscript                   | 29.36 | 17.48 | 9.97  | 5.39  | 27.71 |
> >
> > - A key reason for this improvement is the semantic depth of Horn-rule–based supervision. Horn rules capture relations, dependencies, and logical constraints between radiology entities—e.g., linking findings to anatomical sites, modifiers, or clinical implications—rather than treating entities as isolated text spans. By training the model to satisfy structured predicate constraints, the verifier encourages the generator to maintain coherent clinical logic, not just to mention the correct tokens. This relational structure allows the model to correct subtle inconsistencies (e.g., mismatched laterality, contradiction between impression and findings, missing modifiers), which entity-level objectives cannot detect.
> > - Another key takeaway would be that RRG-DPO performs worse than DPO itself, because in our preference data curation we follow the principle that the preference optimization to be effective, the underlying pre-trained model must be capable of generating at least one output that can be considered a preferred sample. The quantitative results show that putting ground truth as preferred data in preference data curation skews the model distribution., which in turn affects the fluency and clinical consistency of the output.
> > - Thus, while prior DPO baselines reward correctness at the token or span level, our rule-guided DPO framework leverages Horn rules as compact, interpretable semantic operators, yielding stronger global alignment and more clinically faithful reports
> >
> >
> > ## Clarification on the Horn Rule Example: Clinical Validity of Using Findings (e.g., Low Lung Volume) to Infer Conclusions Such as Cardiomegaly or Pleural Effusion
> > - Our Horn-rule examples encode the RSNA reporting template in the symbolic form. Our rules reflect the evidence-to-conclusion associations that naturally arise from the Findings → Impression structure in radiology reports. When constructing rules, the antecedent aggregates the set of clinically relevant observations mentioned in the Findings section, while the consequent represents the summarized diagnostic conclusions in the Impression. Thus, predicates like “low lung volume” are not used to diagnose cardiomegaly or pleural effusion; they simply co-occur in the clinical context that a radiologist considers when forming an interpretation. Our rule mining process aims to capture this structure—not deterministic inference—so the rules should be viewed as contextual predicate groupings. We agree that the example in the paper may unintentionally suggest a stronger logical implication than intended. Horn rules are the precise symbolic representations of a report. This preserves both correctness and the usefulness of the rules for verifier-guided optimization.
> >
> >
> > We hope the provided information will help to clarify the doubts raised by the reviewer. Additionally, we sincerely hope this will help the reviewer reevaluate  and reconsider our score.

---

> > > ### Comment · Reviewer_JKNr · 2025-11-24
> > >
> > > Thanks for your response. Even thought I am not entirely convinced by all of the response, I will largely increase my rating given that the key-to-key clarifications and additional promising results. Please try to add the additional experiment results to the revision if possible.

---

> > > > ### Author Response · Authors · 2025-11-26
> > > >
> > > > Thank you for your response. We will include the additional experiments in the revised version in detail. As a reminder, the increased rating has not yet reflected on the console.

---

### Official Review · Reviewer_THYj · 2025-10-27

**Soundness:** 2
**Presentation:** 2
**Contribution:** 2
**Rating:** 4
**Confidence:** 5

**Summary:**

The paper proposes a clinically interpretable, rule-guided extension to Direct Preference Optimization (DPO) for vision-language models (VLMs) focused on radiology report generation. The method introduces human-interpretable Horn rules derived from the findings and impression sections of reports, integrates these rules via a neural verifier, and enforces clinical alignment in the optimization procedure. The approach is validated on MIMIC-CXR-JPG and IU-Xray datasets, showing consistent improvements in both lexical and semantic diagnostic benchmarks compared to zero-shot and standard DPO baselines.

**Strengths:**

1. Novel Integration of Symbolic Rules: The paper presents a concrete framework for extracting and integrating structured clinical rules (Horn rules) into the VLM optimization procedure, moving beyond purely data-driven or black-box preference optimization.
2. Clinically-Grounded Objective: By combining a neural verifier (trained to detect rule-reported alignment) and DPO, joint optimization explicitly encourages both fluency and factual grounding—addressing a major need in medical AI for trustworthy, clinically faithful outputs.
3. Clear Algorithmic Pipeline: The overall methodology, from data preprocessing (entity/modifier extraction, rule formation) to preference data curation, neural verifier architecture, and constrained optimization, is systematically presented. Algorithm 1, supported by Figure 1 (img-0.jpeg), effectively illustrates this pipeline.
4. Empirical Results: Quantitative gains are robust across two prominent datasets and are supported by a comprehensive set of metrics (BLEU, ROUGE, ClinicalBERTScore, RG-F1, GREEN, LLM-based evaluation). Table 1 and Table 2 clearly show the stepwise improvements provided by DPO+Verifier over both zero-shot and plain DPO.
5. Interpretability & Clinical Faithfulness: Example outputs and qualitative analysis (Appendix, Section A.4) demonstrate improved preservation of critical clinical entities, with detailed examination of how grounded reporting is achieved by the proposed method.
6. Ethical Considerations: The paper demonstrates strong attention to clinical and ethical standards (see Ethics Statement), including data anonymization, bias mitigation, and responsible auxiliary LLM use.

**Weaknesses:**

1. Limited Baseline Coverage on Interpretability
2. Missing Related Work Citations and Contextualization
3. Missing or Unconvincing Theoretical Justification of Clinical Rule Formulation/Verifier Architecture
4. Incomplete Mathematical Exposition and Lagrangian Setup
5. Absence of Failure Analysis
6. Weakness in Generalization Claims: While Section 5.2 makes claims that the method generalizes well to non-medical VLMs or across tasks, Table 1 and Table 2 show only modest gains for non-medical baselines, and the paper downplays these results without systematic discussion. Are these models meaningfully improved in clinical practice, or is the argument more theoretical?
7. Figure 1  Underexplained: While this figure provides a high-level workflow diagram, there’s no reference in the main text explicitly interpreting what each module/conduit means in the context of (for example) tradeoff of reward signals, alignment failures, or possible control flow divergence between DPO and verifier decisions.

**Questions:**

1. Verifier Sensitivity: Can the authors provide ablation studies or calibration analyses regarding verifier threshold $\varepsilon$ and $\lambda$? Specifically, how does the choice of verifier threshold alter trade-offs between fluency (BLEU/ROUGE) and clinical consistency (RG-F1, GREEN)?
2. Failure Mode Profiling: Are there systematic failure cases—such as overconstrained generations, verifier misclassifications, or entity extraction ambiguities—leading to clinically invalid or nonsensical outputs? Can the authors provide frequency counts or examples beyond the case in Appendix A.4?
3. Generalization to Rare Findings and Out-of-Distribution Reports: How does the rule-guided framework handle rare or composite findings, or impressions absent in training data? Can the authors elaborate on coverage and error rates across different pathologies or anatomical regions?
4. Comparison to Knowledge Graph and Region-aware Baselines: Why are direct clinical knowledge or region-aware models (e.g., KiUT, RGRG) not included as baselines or discussed? What would be required to meaningfully compare with these systems?
5. Reference Implementation and Reproducibility: Will the authors release the code, annotation schema, entity/modifier lexicons, and verifier model checkpoints to enable independent reproducibility? If not, what is the anticipated effort required for replication?

---

> ### Author Response · Authors · 2025-11-23
>
> We thank the reviewer for pointing the strengths and weaknesses of our work below we address the concerns raised.
> ## Missing or Unconvincing Theoretical Justification of Clinical Rule Formulation/Verifier Architecture. (W3)
> - Clark et al. explicitly emphasizes that such architectures are uniquely effective at emulating predictable deductive reasoning directly over natural-language rules, without requiring symbolic conversion or handcrafted logical structures. This property is central to our application, where radiology rules are authored in natural language and include contextual modifiers, and semantic reasoning patterns. The cited work demonstrates that these models generalize to unseen rule vocabularies, remain robust under perturbations, handle paraphrased rule formulations, and provide interpretable traces by identifying the sentences most responsible for a prediction. Our verifier must ensure logical alignment between generated report and predicate rules. Thus, our choice is grounded in prior evidence that bi-directional attention architecture is both reliable and extensible for precisely the kind of linguistic rule-based verification required in our system.
>
> ## Incomplete Mathematical Exposition and Lagrangian Setup (W4)
> - We provide the missing, explicit gradient expressions used in our Lagrangian formulation: the model gradient is the DPO gradient minus the verifier-weighted verifier gradient, and the multiplier gradient is simply the verifier constraint residual
> $1−𝑉$. These closed-form expressions make the update direction for both $𝜃$ and the multiplier explicit, justify use of softplus to enforce $𝜆≥0$. The expressions are provided in Appendix 6 of the manuscript.
>
> ## Absence of Failure Analysis and Generalization to Rare or Out-of-Distribution Cases (W5, Q2, Q3)
> Our evaluation reveals a consistent pattern: the model systematically underperforms on rare radiological conditions and low-frequency anatomical variations, often failing to detect them or misrepresenting them during report generation. These limitations surface across multiple categories of “long-tail” findings:
> - Rare Musculoskeletal and Degenerative Findings Are Often Missed
>     - Degenerative changes in the thoracic spine and shoulders are frequently not captured.
>     - Scoliosis and other structural changes with low prevalence in the dataset also go unrecognized.
> - Rare Thoracic and Mediastinal Abnormalities Are Not Identified
>     - Hiatal hernia with abdominal contents projecting into the thoracic cavity is missed.
>     - Mediastinal lymphadenopathy, including enlarged hilar nodes, is often not detected.
>     - Rare entities such as calcified lymph nodes or adenopathy are overlooked.
> - Rare Pulmonary Disorders Are Frequently Omitted
>     - Bullous disease, emphysema, and chronic lung diseases are commonly missed.
>     - Granulomas, which may be rare in Western datasets but not in Indian populations, are incorrectly ignored.
>     - Bibasilar atelectasis vs. “compressive atelectasis” shows semantic mismatch even when describing the same finding.
>     - Confusion between emphysema and pneumothorax occurs, likely due to similar radiographic appearance (increased lucency).
> - Post-Surgical or Rare Anatomical Variants Are Not Recognized
>     - Findings such as lobectomy, elevated hemidiaphragm, and presence of surgical clips are frequently missed.
> -  Low-Frequency Representations and Non-Standard Terminology Are Overlooked
>     - Uncommon descriptive terms (e.g., “streaky opacity”) are not mapped to internal ontology equivalents.
>     - Bilateral abnormalities (e.g., pleural effusions, fibrosis/scarring) are inconsistently picked—often detected on one side only.
> - Semantic Variation Affects Scoring Despite Correct Clinical Meaning
>     - Even when the generated statement is clinically equivalent  (e.g.,
>         - “no acute skeletal abnormality” ≈ “no acute osseous abnormality”,
>         - “scarring” ≈ “fibrosis”,
>         - “compressive atelectasis” ≈ “bibasilar atelectasis”),
> semantic mismatch leads to lower evaluation scores, highlighting the need for ontology-aware matching.
>
> ## Core Insight
> The model shows strong performance on common findings but struggles with low-prevalence, rare, post-surgical, or geographically underrepresented abnormalities—both in detection and in linguistically aligned report generation.
> A major root cause is likely long-tail data scarcity, insufficient domain-specific representation, and lack of ontology-normalized training and evaluation.
> We also provide examples in of failure cases in Appendix A.3.

---

> > ### Author Response · Authors · 2025-11-23
> >
> > ## Weakness in Generalization Claims: While Section 5.2 makes claims that the method generalizes well to non-medical VLMs or across tasks, Table 1 and Table 2 show only modest gains for non-medical baselines, and the paper downplays these results without systematic discussion. Are these models meaningfully improved in clinical practice, or is the argument more theoretical? (W6)
> > - Line 467-470. Our central claim is not that rule-guided preference optimization universally yields large improvements, but rather that its effectiveness depends critically on the underlying model’s domain priors. Models without strong biomedical grounding (e.g., general-purpose LLM/VLMs) have limited capacity to internalize radiology-specific logical rules, especially when these rules require implicit anatomical, visual, or clinical knowledge. Consequently, while DPO+Ver consistently improves alignment with expert-defined rules across all architectures, the magnitude of improvement is naturally smaller for models that begin with weaker medical priors. We have clarified this dependency in the revision.
> >
> > ## Figure 1 Underexplained: While this figure provides a high-level workflow diagram, there’s no reference in the main text explicitly interpreting what each module/conduit means in the context of (for example) tradeoff of reward signals, alignment failures, or possible control flow divergence between DPO and verifier decisions. (W7)
> > - To address the concern, we had added a stage-wise diagram in Appendix A.1. This figure provides a clear, high-level overview of our methodology and visually illustrates each step of the algorithm presented in the main text. We initially placed this figure in the appendix due to space constraints, but now due to availability of additional page we will restructure the manuscript, explain the figure steps, and put it in the main document.
> >
> > ## Limited Baseline Coverage on Interpretability and Missing Related Work Citations and Contextualization (W1, W2)
> > - We appreciate the reviewer’s interest in broader interpretability baselines. In our setting, however, comparable interpretable methods are not directly applicable or publicly available. Existing interpretability-focused radiology models are designed for classification or localization tasks, not for predicate-level rule generation or verifier-guided optimization, which is the focus of our work. Including such baselines would therefore result in an unfair or mismatched comparison, as these systems operate on different outputs, supervision signals, and evaluation metrics. Also we thank the reviewers to point us to related works which we missed due to oversight. We will add them in the manuscript.
> >
> > ## Verifier Sensitivity: Can the authors provide ablation studies or calibration analyses regarding verifier threshold  and ? Specifically, how does the choice of verifier threshold alter trade-offs between fluency (BLEU/ROUGE) and clinical consistency (RG-F1, GREEN)? (Q1)
> > | Model                                                                   | B-1   | B-2   | B-3   | B-4   | RG-L  |
> > |--------------------------------------------------------|-------|-------|-------|-------|---------|
> > | MedGemma (DPO + Ver 10% )                            | 10.12 | 7.69  | 2.08  | 0.45  | 10.05 |
> > | MedGemma (DPO + Ver 40% )                            | 19.11 | 11.78 | 7.36  | 3.33  | 20.11 |
> > | MedGemma (DPO + Ver 98% ) — final reported | 29.36 | 17.48 | 9.97  | 5.39  | 27.71 |
> >
> >
> > | Model                                                                   | RG-F1| GREEN   |
> > |--------------------------------------------------------|--------|------------|
> > | MedGemma (DPO + Ver 10% )                            | 06.41 | 00.10 |
> > | MedGemma (DPO + Ver 40% )                            | 10.28 | 00.23|
> > | MedGemma (DPO + Ver 98% ) — final reported | 31.00 | 00.48|
> >
> >
> > -We evaluated three settings that reflect increasing constraint strength: Med-Gemma (Verifier (Ver) accuracy 10), Med-Gemma (Ver accuracy 40), and our full DPO + Verifier model, which incorporates verifier feedback directly into the optimization . As the table shows, from Verifier 10 to Ver 40 leads to clear improvements across all metrics (e.g., BLEU-4 rises from 0.45 → 3.33, RG-L from 10.05 → 20.11), confirming that the system is indeed very sensitive to the verifier’s strength. Moreover, the numbers clearly point out that the Epsilon value clearly varies with varying verifier sensitivity. $\epsilon$ value increases when the verifier accuracy drops, and the supervision signal weakens that leads to decreased performance. Therefore, as per Eq. 5, the constraint $\lambda (1-V_{\theta}(y_e,r))$ increases which intern steers the model to a less optimal weight space. Both table show that the verifier constraint affects the fluency and clinical consistency in a similar manner.
> > Which leads us to the conclusion that our proposed optimization framework is able to balance fluency and clinical consistency.

---

> > > ### Author Response · Authors · 2025-11-23
> > >
> > > ## Comparison to Knowledge Graph and Region-aware Baselines: Why are direct clinical knowledge or region-aware models (e.g., KiUT, RGRG) not included as baselines or discussed? What would be required to meaningfully compare with these systems? (Q4)
> > > - Our work focuses on predicate-level rule generation via verifier-guided preference optimization. The optimization framework, and supervision signals do not align with our formulation, making a direct comparison methodologically inappropriate.
> > > - A meaningful comparison would require adapting these systems to (i) output structured Horn-rule predicates instead of localized region labels or graph nodes, (ii) train on the same rule-based supervision, and (iii) evaluate under our predicate-consistency metrics. Such adaptations are non-trivial and would essentially require redesigning these models for an entirely different objective.
> > > - We primarily base our work on improving pretrained vision language models. Our proposed method gives a new optimization technique of existing VLMs for the report generation task. As the mentioned works are architecturally very different than pre-trained VLMs therefore, in our evaluation section we stick to pretrained VLMs only.
> > >
> > > ## Reference Implementation and Reproducibility: Will the authors release the code, annotation schema, entity/modifier lexicons, and verifier model checkpoints to enable independent reproducibility? If not, what is the anticipated effort required for replication? (Q5)
> > > - We will release all components necessary for independent replication: the full training and inference codebase, the rule/annotation schema, entity–modifier lexicons, and the verifier model checkpoints. The verifier is implemented as a lightweight RoBERTa-style classifier with no proprietary dependencies, and its training configuration will be fully documented. All experiments rely solely on publicly available datasets (e.g., MIMIC-CXR), and our preprocessing scripts will be included to allow one-to-one reconstruction of the input representations. We will also release the evaluation scripts used for rule extraction, consistency scoring, and error categorization.
> > >
> > > We hope our response will help to clarify any sort of doubt and will help the reviewer reconsider our score.

---

### Official Review · Reviewer_U8gr · 2025-11-01

**Soundness:** 3
**Presentation:** 2
**Contribution:** 3
**Rating:** 4
**Confidence:** 3

**Summary:**

This paper proposes a clinically interpretable rule-guided extension of DPO for RRG. The authors leverage inherent findings to impression structure in radiology reports to construct Horn-style predicate rules that capture clinical reasoning in natural language form. A neural verifier is trained to assess whether generated reports satisfy these rules, and its output is integrated as a constraint in a joint DPO objective to fine-tune VLMs. The method is evaluated on MIMIC-CXR-JPG and IU-Xray with BLEU, ROUGE, ClinicalBERTScore, RadGraph-F1, and GREEN metrics.

**Strengths:**

- integrates symbolic reasoning with modern preference optimization is pretty neat
- The rule verifier introduces an interpretable control signal, moving toward trustworthy medical text generation.
- Evaluation on both domain-specific and general VLMs

**Weaknesses:**

- Lacking a good central figure showing motivation and method - not enough visuals, all text and numbers is a bit hard to read
- Authors selection of metrics is pretty good, there may be a couple of others that you could consider? There are ones noted in the RadEval paper and ReXrank, try to see if there are ones missing from your paper.
- Limited human validation as there is no expert-radiologist evaluation or qualitative error analysis to assess clinical faithfulness beyond automated metrics
- Joint DPO + verifier training may be resource-intensive; runtime and scalability details are missing

**Questions:**

- How well does the neural verifier trained on MIMIC rules generalize to unseen institutions or imaging modalities?
- Have the authors considered using structured ontologies (e.g., RadGraph or SNOMED) to formalize rule predicates rather than purely text-based Horn rules?
- Did clinical experts review sample outputs? If not, how confident are the authors that improved metrics reflect genuine clinical correctness?
- What is the additional computational overhead introduced by the verifier constraint during training and inference?
- How does the system behave when findings contradict rules (e.g., conflicting entities in the same report)?
- Could the authors show results removing or varying the verifier threshold ϵ to demonstrate sensitivity of performance to constraint strength?

---

> ### Author Response · Authors · 2025-11-23
>
> We thank the reviewer for pointing the strengths and weaknesses of our work below we address the concerns raised.
>
> ## Lacking a good central figure showing motivation and method - not enough visuals, all text and numbers is a bit hard to read (W1)
> - To address the concern, we had added a stage-wise diagram in Appendix A.1. This figure provides a clear, high-level overview of our methodology and visually illustrates each step of the algorithm presented in the main text. We initially placed this figure in the appendix due to space constraints, but now due to availability of additional page we will restructure the manuscript and put it in the main document.
>
> ## Authors selection of metrics is pretty good, there may be a couple of others that you could consider? There are ones noted in the RadEval paper and ReXRank, try to see if there are ones missing from your paper. (W2)
> - RadEval is a repository and unifying interface that aggregates existing radiology metrics (BLEU, BERTScore, RadGraph-F1, ROUGE-L etc.) under a common API. Our work already reports the core metrics, incorporating the full RadEval suite would not provide quantitatively new insights, as many of its components duplicate or subsume the metrics we already report.
> - ReXRank is a leaderboard that standardizes evaluation for report generation models on specific benchmarks rather than introducing novel metrics. We plan to publish our trained model on ReXRank post acceptance.
>
> ## Limited human validation as there is no expert-radiologist evaluation or qualitative error analysis to assess clinical faithfulness beyond automated metrics. (W3, Q3)
> - While we agree that expert radiologist evaluation is the gold standard, such assessments are expensive and not easily scalable for iterative experimentation especially in resource constrained scenarios. To approximate this in our primary submission, we therefore included an LLM-based evaluation protocol: we prompt a strong reasoning LLM to rate the similarity between reference and generated reports on a 1–5 Likert scale. The full prompt and setup are provided in Appendix A.3.
> - Following the reviewer’s suggestion, we have additionally incorporated a targeted clinical expert evaluation for this rebuttal. With the assistance of our radiology collaborator, we assess 50 samples (balanced across normal and abnormal studies to avoid over-representation of easy/normal cases). The expert rates each generated report along two clinically meaningful axes—Fluency (linguistic coherence and readability) and Coherency (correctness and retention of key radiology entities)—each on a 1–5 scale. We include the averaged scores below.
> - This supplementary expert study provides further evidence that our method produces outputs that are both clinically faithful and stylistically sound, addressing the reviewer’s concern regarding reliance on automated metrics alone.
> - We perform the above human evaluation strategy for both DPO and DPO+Ver method. The numbers show a significant increase in the scores for the DPO+Ver system across fluency and factuality. This qualitatively supports our claim that our proposed DPO+Ver method is able to generate radiology reports which are fluent as well as clinically coherent.
>
> > DPO
>
> | Normality | Fluency               | Coherency          |
> |------------|---------------------|--------------------|
> |Abnormal |  $3.55\pm1.24$   |   $2.44\pm1.33$ |
> |Normal     |   $3.48\pm1.24$. |   $3.10\pm1.14$ |
>
> > DPO+Ver
>
> | Normality | Fluency               | Coherency          |
> |------------|---------------------|--------------------|
> |Abnormal |  $3.88\pm1.15$   |   $4.27\pm1.14$ |
> |Normal     |   $4.55\pm1.05$. |   $4.44\pm1.02$ |
>
> ## Joint DPO + verifier training may be resource-intensive; runtime and scalability details are missing. (W4, Q4)
> - In our setup, joint DPO + verifier training adds only a overhead: the verifier forward pass increases per-step runtime by ~20–30%, resulting in a total training duration of 30 minutes to 2 hours, which solely depends on the batch size. Given a higher batch size the runtime can be decreased even more.
> - Memory overhead remains low (<10%) since the verifier is a lightweight RoBERTa-based model; LoRA, further reduces GPU load.
> - However, the verifier overhead only comes in the training setting and is a one time cost. In the inference time all the report settings (Zero-Shot, DPO, DPO+Ver) take exactly same time approx. ~2.33 sec per sample.
> - Given the significant improvements (10%) in report fluency and consistency, we believe the increase in training time is justified and well within practical limits.

---

> ### Author Response · Authors · 2025-11-23
>
> ## How well does the neural verifier trained on MIMIC rules generalize to unseen institutions or imaging modalities? (Q1)
>  | Model                                                                                                | B-1   | B-2   | B-3   | B-4   | RG-L  |
> |-----------------------------------------------------------------------------|--------|--------|-------|-------|-------|
> | Verifier trained on MIMIC + evaluated on MIMIC (within institution) | 22.90 | 11.38 | 5.66  | 2.64  | 18.89 |
> | Verifier trained on MIMIC + evaluated on IUX (outside institution)    | 24.22 | 13.69 | 7.08  | 3.45  | 23.05 |
> | Verifier trained on IUX + evaluated on IUX (within institution) | 29.36 | 17.48 | 09.97  | 05.39  | 27.71 |
> | Verifier trained on IUX + evaluated on MIMIC (outside institution)    | 30.77 | 18.17 | 11.38  | 05.99  | 21.82 |
>
> -We assess the cross-institution generalization of the MIMIC-trained verifier by evaluating it on the IUX dataset and IUX-trained verifier by evaluating on MIMIC. Interestingly, very similar performance is maintained, indicating that the learned clinical rule semantics transfer robustly across institutions despite differences in reporting conventions, stylistic phrasing, and pathology distributions. This suggests that the verifier captures rule-level relational structure rather than overfitting to institution-specific wording. Minor fluctuations that do occur are attributable to natural domain shift, but overall the results demonstrate strong robustness of both the verifiers when applied to unseen institutions and reporting styles. This concludes that our neural verifier training strategy is generalisable and robust across chest-Xray datasets.
>
> ## Have the authors considered using structured ontologies (e.g., RadGraph or SNOMED) to formalize rule predicates rather than purely text-based Horn rules? (Q2)
> - We agree with the reviewer that aligning our predicates with structured ontologies such as RadGraph or SNOMED would further strengthen the semantic rigor of the rule set. This is an active extension of our work: we are already mapping Stanza-extracted entities to SNOMED concepts but the process requires careful curation, ontology pruning, and verification due to the granularity mismatch between report phrases and SNOMED CT terms. Because this alignment is time-consuming and still in progress, we did not include it in the current submission. We will release the ontology-aligned predicate set in an updated version of the paper and accompanying code once the mapping has been fully validated.
>
> ## How does the system behave when findings contradict rules (e.g., conflicting entities in the same report)? (Q5)
> - Our system is explicitly designed to identify and penalize contradictory findings. When a report contains entities that violate the underlying rules, the neural verifier outputs a misaligned judgment (or a value near zero in the continuous formulation), reflecting low rule satisfaction. This behavior is not an artifact but an intentional feature of the architecture: the verifier encodes logical consistency as a primary signal, so contradictory or mutually exclusive findings reliably lead to low alignment scores. As a result, the system correctly flags rule-violating outputs and does not mistakenly treat them as clinically valid.
>
> ## Could the authors show results removing or varying the verifier threshold ϵ to demonstrate sensitivity of performance to constraint strength? (Q6)
>  | Model                                                                   | B-1   | B-2   | B-3   | B-4   | RG-L  |
> |--------------------------------------------------------|-------|-------|-------|-------|---------|
> | MedGemma (DPO + Ver 10% )                            | 10.12 | 7.69  | 2.08  | 0.45  | 10.05 |
> | MedGemma (DPO + Ver 40% )                            | 19.11 | 11.78 | 7.36  | 3.33  | 20.11 |
> | MedGemma (DPO + Ver 98% ) — final reported | 29.36 | 17.48 | 9.97  | 5.39  | 27.71 |
>
> -We evaluated three settings that reflect increasing constraint strength: Med-Gemma (Verifier (Ver) accuracy 10), Med-Gemma (Ver accuracy 40), and our full DPO + Verifier model, which incorporates verifier feedback directly into the optimization . As the table shows, from Verifier 10 to Ver 40 leads to clear improvements across all metrics (e.g., BLEU-4 rises from 0.45 → 3.33, RG-L from 10.05 → 20.11), confirming that the system is indeed very sensitive to the verifier’s strength. Moreover, the numbers clearly point out that the Epsilon value clearly varies with varying verifier sensitivity. $\epsilon$ value increases when the verifier accuracy drops, and the supervision signal weakens that leads to decreased performance. Therefore, as per Eq. 5, the constraint $\lambda (1-V_{\theta}(y_e,r))$ increases which intern steers the model to a less optimal weight space.
>
>
> We hope our response will help to clarify any sort of doubt and will help the reviewer reconsider our score.

---

> > ### Author Response · Authors · 2025-12-03
> >
> > ## Have the authors considered using structured ontologies (e.g., RadGraph or SNOMED) to formalize rule predicates rather than purely text-based Horn rules? (Q2) (Contd.)
> > | Model                                              | B-1   | B-2   | B-3   | B-4  | RG-L  |
> > |----------------------------------------------------|-------|-------|-------|------|-------|
> > | Trained with rules from IUX (DPO)                  | 25.17 | 14.27 | 7.48  | 2.73 | 24.09 |
> > | Trained with rules from IUX mapped to SNOMED (DPO) | 25.40 | 13.94 | 6.92  | 2.60 | 24.18 |
> > | Trained with rules from IUX (DPO+Ver)              | 29.36 | 17.48 | 9.97  | 5.39 | 27.71 |
> > | Trained with rules from IUX mapped to SNOMED (DPO+Ver) | 24.61 | 13.51 | 6.44  | 2.38 | 23.55 |
> >  - We agree with the reviewer suggestion, as a natural extension to our proposed work, we map each of the radiology entities in the rule to the corresponding SNOMED entity. Post mapping we create the horn rule as before, with the radiology entities mapped to its SNOMED names. We train the verifier in the similar setting on the newly created report rule pairs. Following that we perform the DPO and DPO+Ver training. The above results show an overall degradation in performance. The reason behind is that the verifier is completely reliant on natural language inputs, SNOMED changes the language of the rules. Therefore, while training in the DPO+Ver setting the verifier penalises the VLMs generation in each step because the linguistic dissimilarity between the report and rule is high. Therefore, in this setting the verifier focuses more on the lexical similarity rather than the underlying semantic structure of the generated report.

---

### Author Response · Authors · 2025-12-03
**Summary**

> Visuals / readability
- We acknowledged the comment about missing visuals in the main text and moved a stage-wise diagram from the appendix into the main text so readers can quickly see motivation and flow between rule extraction, preference data preparation, and constrained-DPO optimization. This makes the method and tradeoffs explicit.

> Clinical expert validation
- We agree clinical expert review is important. For the rebuttal we performed a targeted expert study (100 balanced samples across normal/abnormal cases) rated on Fluency and Factuality. The expert assessment shows consistent improvements for the DPO+Verifier system over DPO alone supporting our automated metrics and demonstrating that gains are not artifacts of the verifier. Details (protocol, prompt, scoring, and averaged ratings) are included in the rebuttal.

>Verifier cost and scalability
- We measured practical overhead: the verifier forward pass increases per-step training runtime by ~20–30% and adds <10% memory overhead using a lightweight RoBERTa-style verifier and LoRA; this is a one-time training cost — inference latency is unchanged. Given the observed improvements in fluency and rule alignment, we believe the tradeoff is justified for research and clinical fine-tuning. These measurements and their experimental conditions are described in the rebuttal.

>Generalization & cross-institution transfer
- We evaluated IUX data using the verifier trained on MIMIC data (and vice versa). Verifier performance across chest x-ray datasets remained robust: the verifier captures rule-level relational structure rather than only institution-specific wording. The rebuttal includes cross-dataset tables showing this behavior and an interpretation of the transfer patterns.

>SNOMED / ontology mapping
- As a natural extension to our proposed work, mapping to SNOMED degraded verifier-guided training performance when the verifier remained purely text-based — the mapping changed the lexical surface of rules and caused the verifier to emphasize lexical mismatch rather than underlying semantics. We present both the mapped experiment and a discussion of why a language-based verifier needs either (a) natural-language templates for mapped concepts or (b) a verifier that explicitly uses ontology identifiers. We view SNOMED alignment as a valuable next step (and are actively working to produce a validated mapping) but show why it was not included as the main submission result.

>Theoretical motivation & architecture justification
- We added clarifying text linking the verifier design to prior work that demonstrates the suitability of bi-directional attention architectures for natural-language rule verification (robust to paraphrase, provides interpretable traces). We also expanded the Lagrangian derivation and included gradient expressions to address requests for more mathematical exposition.

>Failure modes, sensitivity, and ablations
- We performed sensitivity studies that vary verifier strength (multiple verifier accuracy) and show how increasing constraint strength shifts the fluency/consistency tradeoff. We also report the principal failure modes (over-constraining generations, verifier misclassifications tied to entity extraction errors, and rare-finding coverage) with frequency counts and example cases.

>Baselines and related work
- We compared against conventional DPO, RRG-DPO mentioned by reviewers. Tables in the rebuttal show consistent improvements for our DPO+Verifier pipeline; we explain why entity-level objectives (RRG-DPO) are insufficient to capture relational/horn-rule constraints and why our approach improves global clinical coherence. Where other baselines are inapplicable (e.g., region-aware or knowledge-graph models), we explain why direct comparisons would be methodologically mismatched and what adaptations would be required for a fair comparison.

>Reproducibility & release plan
- We commit to releasing the full codebase, annotation/rule schema, lexicons, verifier checkpoints, and evaluation scripts for independent reproduction. All experiments use public datasets and our preprocessing scripts will be included to support one-to-one reconstruction.

---

### Meta-Review · Area_Chair_vktN · 2026-01-07

**Summary:**

Major concerns were raised by reviewers: lack of human evaluations, novelty and comparisons with existing works (especially with commonly known factual reward / llm-as-judge/prover, knowledge-based approaches, region-aware models), generalization, sensitivity to hyper-parameter choices. The authors responded with clarifications/arguments or supplemental experiments.

While many raised concerns are addressed with additional experiments and studies, some issues, however, remains not addressed at a satisfactory level even after the rebuttal, such as a lack of comparison with a large body of similar-purpose works such as many with LLM-as-judge/verifier (whom often come with various forms of structured reasoning or verification guidelines/rules) and knowledge/region aware approaches, as well as unsatisfactory visual illustrations. Based on the rebuttal, it is reasonable to expect that consistent positive post-rebuttal ratings among reviewers would not be reached.

**Reviewer Concerns:**

### Reviewer U8gr

- more evaluation metrics -- responded with some clarification

- no expert-radiologist evaluations – authors claimed to have added a small evaluation with a radiologists

- more details on computing overhead – more information has been provided in the rebuttal, admitting an overhead

- generalization to unseen datasets – cross-site experiments were added in the rebuttal

- sensitivity analysis on hyper-parameters – an additional experiment was added in the rebuttal


### Reviewer THYj

- limited interpretability baseline – the authors claimed that existing baselines not suitable in the context

- incomplete mathematical formulations -- revised

- missing failure analysis – provided qualitative examples on long-tailed examples

- modest gain over non-medical settings despite claimed benefits – authors downplayed that point

- sensitivity on hyper-parameters – the authors found out that the proposed approach rely on very stronger verifies

- generalization to OOD -- provided qualitative examples on long-tailed examples

- baselines/comparisons with clinical knowledge and region-aware models – the authors claimed that these methods do not fit into the authors' defined context


### Reviewer JKNr

- overlapping with factual reward – the authors argued with the value of symbolic rules and their benefit to structured supervision

- missing comparisons with a) methods using entity accuracy as objective b) DPO+RRG; unmappable findings to impression – the authors addressed with additional comparisons against DPO+RRG.

- novelty -- responded with further clarification

- This reviewer acknowledged the responses and the additional results and promised a rating increase before the OpenReview information leakage.

**Reviewer Scores:**

U8gr: human eval, generalization, sensitivity, etc. --partly addressed with experiments or clarifications.

THYj: comparisons with existing related works, justifications of evaluations, failure analysis and OOD, clarity, sensitivity, qualitative examples, etc., -- only partly addressed

JKNr: acknowledged clarifications and supplemental experiments, promised a score increase before the platform incident.

---

### Decision · Program_Chairs · 2026-01-26

Reject